# Pollution Analysis and Health Implications of Heavy Metals under Different Urban Soil Types in a Semi-Arid Environment

Salar Rezapour [1,*], Mehri Azizi [1] and Amin Nouri [2]

1   Soil Science Department, Urmia University, P.O. Box 165, Urmia 57134, Iran;
    ag581650parsianinsurance@gmail.com
2   Hermiston Agricultural Research and Extension Center, Oregon State University, Hermiston, OR 97838, USA;
    amin.nouri.g@gmail.com
*   Correspondence: s.rezapour@urmia.ac.ir or s_rezapour2000@yahoo.com; Fax: +98-44-32779558

**Abstract:** A fundamental requirement for the effective prevention and management of soil contamination involves the determination of heavy metal contamination levels and the assessment of associated health risks for human populations. In this study, an analysis was conducted to evaluate the pollution levels and health risks associated with heavy metals in urban soils, specifically focusing on four distinct soil types, namely Calcisols, Cambisols, Fluvisols, and Regosols. The mean values of five heavy metals (Zn, Cu, Cd, Pb, and Ni), some soil pollution indices, and human health risk indices were determined. Pollution indices including the integrated Newerow pollution index (PIN), single pollution index (PI), and pollution load index (PLI) showed a moderate pollution class in most soil samples. The non-carcinogen risk index of elements (HI) in each exposure pathway and the total of the exposure pathways (THI) was <1 for three different population groups (children, adult females, and adult males) and in all soils. This shows the lack of non-cancerous risk for local residents in the study site. The variations in HI and THI for the three population groups and three different exposure pathways was in the order of adult males > adult females > children and ingestion > dermal contact > inhalation. The carcinogenicity risk (CR) of Cd, Pb, and Ni through exposure by ingestion was $>1 \times 10^{-4}$ for children in all soils, meaning that soil ingestion is hazardous to children in the study region. For all three population groups and all soil types, Pb was most effective in HI and THI, whereas Cd had the highest carcinogenicity potential.

**Keywords:** carcinogenic risk; children; non-carcinogenic risk; soil pollution; soil heavy metals; urban soil

## 1. Introduction

Urban and peri-urban regions represent crucially dense locations worldwide, owing to their extensive provision of services, facilities, industrial establishments, and economic operations. The United Nations' statistical data support this assertion [1]. Approximately 56% of the global population resided in urban and peri-urban areas in 2014, with projections indicating a surge to 69% by the year 2050. This burgeoning urban and peri-urban populace is anticipated to exacerbate human encroachment upon natural resources, particularly soils, leading to significant alterations in their physical, chemical, and biological characteristics. The soils found within these regions, heavily impacted by human endeavors and urban waste, exhibit notable distinctions from their natural counterparts, resulting in widespread perturbations across numerous attributes primarily influenced by the industrial, commercial, and service activities of individuals. At a global level, urbanization accounts for approximately 80% of urban and industrial waste, which encompasses nearly 2% of the Earth's land area [2]. The ecosystems within urban and peri-urban regions can be characterized as intricate networks comprising a multitude of elements generated through human and natural processes. Specifically, urban soils pertain to the terrestrial substrates

found in peri-urban locales, consisting of diverse constituents predominantly shaped by human endeavors, and thus distinct from agricultural and forest soils [3].

In general, an array of human activities in peri-urban regions are implicated in the introduction of various chemical and organic pollutants, as well as elements, into the soil. Among these pollutants, heavy metal contamination has received particular emphasis. A study conducted in Siena, located in Central Italy, demonstrated that the levels of heavy metals in soils followed the pattern of non-urban soil < green-urban soil < urban soil [4]. Ding and Hu [5] revealed that cadmium (Cd) and arsenic (As) in the urban soils of Nanjing, China, posed a high ecological risk. Soil is the most important natural substance to absorb and maintain heavy metals and is a natural buffer to control heavy metals and transfer them to the atmosphere, hydrosphere, and biosphere. Heavy metals are, on the other hand, stable and indecomposable in soil, and soils have a limited capacity to absorb and maintain them, beyond which they may be emitted into the soil–water–plant–animal–human ecosystem and pose a serious threat to human health [6,7]. They can also find their way into the human body through ingestion, inhalation, and dermal contact and cause various cancerous and non-cancerous diseases, irrespective of age or gender. The heavy metals not metabolized in the body and can accumulate in tissues such as fats, muscles, bones, and joints, and this can cause different diseases. Some of these elements have no pollutant threshold—that is, they are harmful at any concentration and can entail incurable and fatal diseases in the long run by accumulation in the tissues of living organisms. Lead (Pb) and cadmium (Cd) are two of the most important elements whose accumulation in human tissues may be deeply harmful to the neural and enzymatic systems and may cause lung cancer and bone-breaking [8].

In urban areas, heavy metals can enter soil ecosystems from various sources induced by anthropogenic activities, such as the facilities of metal melting, plating, dying, vehicles, mechanical tools, agronomic activities, the wastewater of utilities, the service sector, battery manufacturing, and so on. Huang et al. [9] and Hu [10] reported that traffic and agricultural activities were the main sources of heavy metals in some peri-urban soils in China. In contrast, Walraven et al. [11] in the Netherlands, Gąsiorek et al. [12] in Poland, and Baltas et al. [13] in Turkey reported a series of industrial activities and atmospheric precipitation as the main sources of urban heavy metals. On the other hand, the behavior and characteristics of urban and peri-urban soils are very complicated as they are influenced by a set of natural and artificial factors and processes and these characteristics can dictate the behavior of heavy metals and their changes [2,14,15]. Accordingly, an assessment of the pollution degree and health risk of heavy metals in the soils of these regions has created a new challenge for researchers in most parts of the world.

Previous research has acknowledged the individual influence of agricultural and service–industrial activities on pollution levels and the presence of heavy elements in urban and suburban soils [16,17]. However, the current body of evidence lacks comprehensive data regarding the collective impact of these activities (agricultural and service–industrial activities) on the characteristics, sources, and health risk assessment of heavy metals in soils, particularly in regions exhibiting diverse soil types and affected by urban activities.

The research site chosen for this study is situated in Urmia, a region in northwestern Iran characterized by a semi-arid climate. This location accommodates a variety of small and medium-sized service and industrial establishments, including food processors, cold storage facilities, dying facilities, and metal plating facilities, among others. The effluents and waste products generated by these industries are directly and/or indirectly discharged into the agricultural lands within the region. Additionally, the outskirts of the city of Urmia serve as a disposal site for most of the city's waste, often situated along the periphery of the farms in the area. Moreover, the region is subjected to the emissions of pollutants from the heavy traffic that traverses the outer belt of the city. Past studies [7,10,18] have demonstrated that any single factor mentioned above can result in a substantial release of heavy metals into soil. However, limited data exist regarding the magnitude of pollution and the associated health risks in agricultural lands situated in urban and peri-urban areas

within arid and semi-arid regions, particularly those with diverse soil types including calcareous soils.

The main objectives of the present study are as follows: (1) to determine the quantity, origin, and pollution index of Ni, Pb, Cd, Cu, and Zn in urban soils, (2) to analyze the health risk of heavy metals to humans (different genders and ages) through the assessment of exposure, including ingestion, dermal contact, and inhalation in urban soils, and (3) to assess the impact of different urban soil types on the quantity, pollution index, and health risk of heavy metals.

## 2. Materials and Methods

### 2.1. Study Site and Fieldwork

The study site was located in the east of the city of Urmia in Western Azerbaijan province, in the northwest of Iran, covering an area between the longitudes of 45°07′20″ and 45°07′38″ E and the latitudes of 37°31′44″ and 37°34′58″ N (Figure 1). The altitude of the study site is 1285–1330 m from sea level, its mean annual precipitation is 338 mm, and it has a semi-arid Mediterranean climate. The parent material of the soils is young alluvium and its oldest geological structures are from the Quaternary. The main land-uses are agriculture and horticulture, chiefly producing wheat, corn, sunflower, and vegetables on farms and grapes and potatoes in orchards. The irrigation water is supplied from the surface water leaving Urmia and from the deep well water.

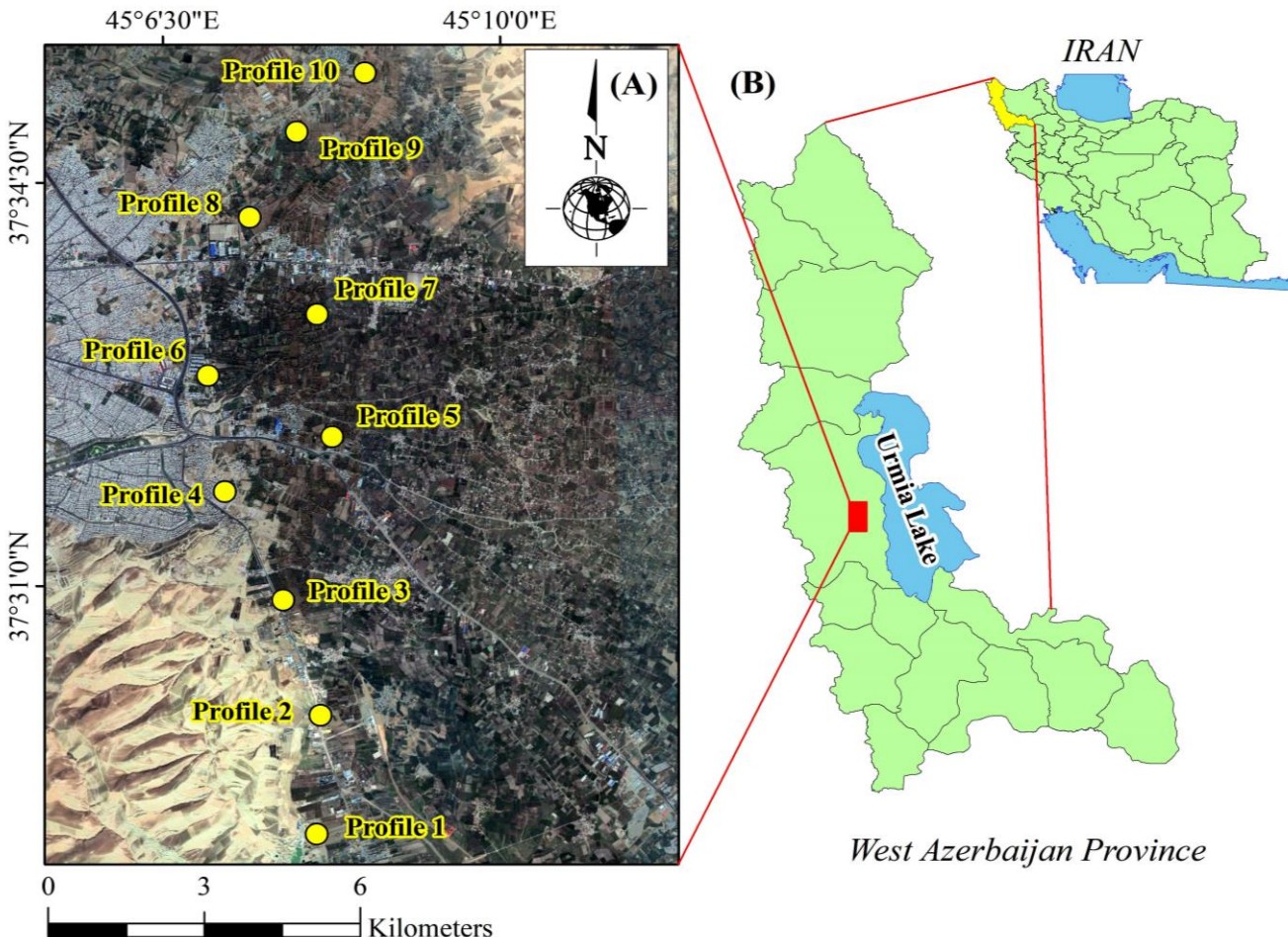

**Figure 1.** Location of the study area. The map of individual study soil profiles (**A**) and location of the study region (**B**).

In the field operation, a total of 10 soil profiles were dug along a 5 km transect. Then, all soil profiles were described and sampled [19]. The soil samples were taken from all

diagnostic horizons of each profile. After determining the morphological characteristics of soil profiles and conducting a physicochemical analysis, the profiles were classified using the World Reference Base (WRB) system [19]. Accordingly, the soils were classified into four main reference groups, including Calcisols (Cal), Cambisols (Cam), Fluvisols (Flu), and Regosols (Reg). Each soil profile was regarded as a central point. Then, composite samples were taken from the soil surface at spots 50–100 m away from these central points in the north–south and east–west directions. Each composite soil sample (0–50 cm depth), which was a mixture of five subsamples, was analyzed in three replications. All soil profiles were characterized by their medium—heavy texture, and a high pH and calcium carbonate.

*2.2. Laboratory Analyses*

The soil samples were prepared, air-dried, and screened through a 2 mm sieve. They were, then, subjected to typical physicochemical analyses. Soil particle size distribution was determined using the hydrometer method [20]. Soil pH and electrical conductivity (EC) were measured using the saturated paste (1:5 soils to 0.01 M $CaCl_2$) and saturated extract methods, respectively [6]. Organic carbon was determined using a wet oxidation technique [21]. Calcium carbonate equivalent (CCE) was estimated after treating the soil with acid (1N HCl) and titration with 1N NaOH [22]. Cation exchange capacity (CEC) was measured using sodium acetate (1 M NaOAc) at pH 8.2 [23]. The elements Zn, Cu, Cd, Pb, and Ni were extracted by digestion in concentrated nitric acid [24]. Based on the method, 10 $cm^3$ of concentrated $HNO_3$ were added to 2 g of soil and heated for 15 min at 95 °C, followed by the addition of 2 $cm^3$ of deionized water and 3 $cm^3$ of 30% $H_2O_2$. The concentrations of these five elements in the extracts were measured via Shimadzu-6300 atomic absorption spectrometry (Shimadzu Corporation, Kyoto, Japan). The highly pure laboratory material of the Merck Group was used to calibrate the d sentences evice. To control the quality of the analyses, some standard samples were prepared, and their concentrations were determined along with the main samples. Their recovery rate was in the range of 94–103%. To validate the measurements of element contents, they were repeated for 20% of the samples, which showed an error of <5%. The analyses of $SiO_2$, $Al_2O_3$, and $Fe_2O_3$ were carried out via X-ray fluorescence (XRF) spectrometry (PANalytical CubiX, WD) from melted-powder pellets of the samples.

*2.3. Soil Pollution and Risk Investigation Indices*

The pollution and risk investigation indices were used as follows [25]:

- Single factor pollution index (PI)

$$PI = \frac{C_i}{S_i} \tag{1}$$

- Nemerow pollution index (PIN)

$$PIN = \sqrt{\frac{\left(\frac{1}{n}\sum_{i=1}^{n} PI_n\right)^2 + [\max(PI)]^2}{2}} \tag{2}$$

In these two equations, $C_i$ is the concentration of the element i, $S_i$ is the background value of the metal i [26], max(PI) is the maximum value of the single factor pollution index, n is the number of the examined metals, and i is the ith value of the metals.

- Pollution load index (PLI) [27]

$$PLI = (PI_1 \times PI_2 \times PI_3 \dots PI_n)^{\frac{1}{n}} \tag{3}$$

- Human health risk assessment indices

Non-carcinogenic and carcinogenic health risk to humans via ingestion, dermal contact, and inhalation was evaluated by calculating the average daily dose (ADD), hazard quotients (HQ), hazard index (HI), and carcinogenic risk (CR) as follows [8,28]:

$$\text{ADD}_{\text{ing}} = \text{C} \times \frac{\text{IR}_{\text{ing}} \times \text{EF} \times \text{ED}}{\text{BW} \times \text{AT}} \times 10^{-6} \tag{4}$$

$$\text{ADD}_{\text{inh}} = \text{C} \times \frac{\text{IR}_{\text{inh}} \times \text{EF} \times \text{ED}}{\text{PEF} \times \text{BW} \times \text{AT}} \tag{5}$$

$$\text{ADD}_{\text{dermal}} = \text{C} \times \frac{\text{SA} \times \text{AF} \times \text{ABS} \times \text{EF} \times \text{ED}}{\text{BW} \times \text{AT}} \times 10^{-6} \tag{6}$$

$$\text{HI} = \sum \text{HQ}_{\text{I}} = \sum \frac{\text{ADD}_{\text{i}}}{\text{RfD}_{\text{i}}} \tag{7}$$

$$\text{THI} = \sum \text{HI} = \sum_{j=1}^{N} \frac{\text{ADD}_{\text{ing}}^{j}}{\text{RfD}_{\text{ing}}^{j}} + \sum_{j=1}^{N} \frac{\text{ADD}_{\text{inh}}^{j}}{\text{RfD}_{\text{inh}}^{j}} + \sum_{j=1}^{N} \frac{\text{ADD}_{\text{dermal}}^{j}}{\text{RfD}_{\text{dermal}}^{j}} \tag{8}$$

$$\text{CR} = \text{ADD}_{\text{i}} \times \text{SF} \tag{9}$$

$$\text{TCR} = \sum_{j=1}^{N} (\text{ADD}_{\text{ing}}^{i} + \text{SF}_{\text{ing}}^{i}) + \sum_{j=1}^{N} (\text{ADD}_{\text{inh}}^{i} + \text{SF}_{\text{inh}}^{i}) + \sum_{j=1}^{N} (\text{ADD}_{\text{dermal}}^{i} + \text{SF}_{\text{dermal}}^{i}) \tag{10}$$

in which $\text{ADD}_{\text{ing}}$, $\text{ADD}_{\text{inh}}$, and $\text{ADD}_{\text{dermal}}$ are ADD from soil ingestion, inhalation, and dermal contact, respectively (mg element $\text{kg}^{-1}$ bodyweight $\text{day}^{-1}$), C is the heavy metal concentration in soil (mg $\text{kg}^{-1}$), THI represents the total exposure hazard index, SF is the carcinogenicity slope factor (mg (kg day)$^{-1}$), and TCR denotes the total carcinogenic risk index. The definitions and values of factors to estimate human health risk are illustrated in Table 1.

**Table 1.** The description and reference values of basic parameters for risk assessment models.

| Parameter | Definition | Unit | Child | Male | Female |
|---|---|---|---|---|---|
| $\text{IR}_{\text{ing}}$ | Ingestion rate | mg $\text{d}^{-1}$ | 200 | 100 | 100 |
| EF | Exposure frequency | day $\text{year}^{-1}$ | 350 | 250 | 250 |
| ED | Exposure duration | year | 6 | 25 | 25 |
| BW | Average body weight | kg | 15 | 68 | 58 |
| AT | Average life span for heavy metal | - | ED × 365 (2190) | ED × 365 (9125) | 9125 |
| $\text{IR}_{\text{inh}}$ | Inhalation rate | $\text{m}^3 \text{ d}^{-1}$ | 7.6 | 20 | 20 |
| PEF | Particulate emission factor | $\text{m}^3 \text{ kg}^{-1}$ | $1.36 \times 10^9$ | $1.36 \times 10^9$ | $1.36 \times 10^9$ |
| SA | Exposed skin area | $\text{cm}^2$ | 2699 | 3950 | 3950 |
| AF | Skin adhesive factor | mg $\text{cm}^{-2} \text{ d}^{-1}$ | 0.2 | 0.07 | 0.07 |
| ABS | Skin absorption factor | - | $1 \times 10^{-3}$ | $1 \times 10^{-3}$ | $1 \times 10^{-3}$ |

*2.4. Data Analysis*

The SPSS 16 software package was used to calculate all statistical parameters, compare the means of the parameters, and draw the graphs (SPSS Inc., Chicago, IL, USA). The same statistical package was used to analyze correlations and perform principal component analysis (PCA) to detect the origin of the elements. The Kaiser–Meyer–Olkin (KMO) test and Bartlett's test of sphericity were used to assess the capability of PCA to accurately sort down the soil attributes for factor analysis. The high value of KMO (0.635) and the

significance level of Bartlett's test of sphericity ($p < 0.000$) showed that that factor analysis was adequate for our data. The means of different indices of soil pollution and human health were compared between the different soil types using one-way analysis of variance. Least squared means were compared using Duncan's test at the $p < 0.05$ confidence level.

## 3. Results and Discussion

### 3.1. Soil Characteristics

Some physicochemical characteristics of the studied soils and their statistical parameters are presented in Table 2. The clay and sand fraction of the soils was in the range of 2.5–71% (an average of 34.8%). About 33% of the soil samples exhibited a texture class of clay and 26% showed a texture class of loam–clay, implying the heavy texture of most samples. The ranges of pH (Min = 7.36, Max = 8.06, Mean = 7.72) and calcium carbonate equivalent (Min = 4.5%, Max = 49.1%, Mean = 23.3%) showed that the soils in the study site were alkaline and calcareous. Salinity (EC) was less than 2 dS m$^{-1}$ in over 95% of the samples. Therefore, the majority of soils were not saline (EC < 4 dSm$^{-1}$). Soil organic matter content was in the range of 0.21–5.24%, with an average of 1.98%, implying a very poor to very strong organic matter content of the soils [29,30]. Cation exchange capacity (CEC) ranged from 5.1 to 42.6 Cmol kg$^{-1}$, with an average of 22.8 Cmol kg$^{-1}$, reflecting a moderate CEC class [29].

**Table 2.** Summary statistics of selected physicochemical properties of the soils.

| Soil Parameter | Calcisols (N = 16) | | | |
|---|---|---|---|---|
| | Min | Max | Mean $\pm$ SD | CV (%) |
| Clay (%) | 17.5 | 43.5 | 29.1 $\pm$ 8.4 | 28.9 |
| Silt (%) | 34.0 | 54.5 | 42.4 $\pm$ 6.1 | 14.3 |
| Sand(%) | 12.0 | 46.0 | 27.8 $\pm$ 11.6 | 41.7 |
| CEC (cmol kg$^{-1}$) | 19.1 | 25.7 | 23.8 $\pm$ 1.8 | 7.5 |
| pH | 7.4 | 8.1 | 7.8 $\pm$ 0.2 | 2.5 |
| EC (dS m$^{-1}$) | 0.7 | 1.0 | 0.8 $\pm$ 0.2 | 19.5 |
| OM (%) | 0.6 | 3.1 | 2.0 $\pm$ 1.1 | 35.3 |
| CCE (%) | 14.5 | 34.0 | 27.6 $\pm$ 5.4 | 19.4 |
| | Cambisols (10) | | | |
| Clay (%) | 44.0 | 50.0 | 47.1 $\pm$ 2.5 | 5.2 |
| Silt (%) | 31.5 | 34.0 | 32.6 $\pm$ 1.1 | 3.2 |
| Sand(%) | 17.0 | 24.0 | 20.3 $\pm$ 2.7 | 13.1 |
| CEC (cmol kg$^{-1}$) | 19.8 | 22.1 | 21.1 $\pm$ 0.9 | 4.2 |
| pH | 7.6 | 7.8 | 7.7 $\pm$ 0.1 | 1.3 |
| EC (dS m$^{-1}$) | 0.62 | 0.75 | 0.7 $\pm$ 0.04 | 6.2 |
| OM (%) | 3.6 | 1.3 | 2.6 $\pm$ 0.3 | 25.8 |
| CCE (%) | 4.5 | 13.0 | 10.6 $\pm$ 3.8 | 35.6 |
| | Fluvisols (12) | | | |
| Clay (%) | 24.0 | 46.0 | 33.8 $\pm$ 8.6 | 25.5 |
| Silt (%) | 26.0 | 46.5 | 35.5 $\pm$ 6.9 | 19.6 |
| Sand(%) | 25.0 | 40.0 | 30.7 $\pm$ 4.4 | 14.3 |
| CEC (cmol kg$^{-1}$) | 21.0 | 31.2 | 25.0 $\pm$ 3.3 | 13.1 |
| pH | 7.4 | 8.0 | 7.7 $\pm$ 0.2 | 2.8 |
| EC (dS m$^{-1}$) | 0.7 | 1.2 | 0.8 $\pm$ 0.2 | 20.9 |
| OM (%) | 1.3 | 5.2 | 3.0 $\pm$ 0.7 | 27.8 |
| CCE (%) | 10.0 | 33.0 | 21.1 $\pm$ 7.7 | 36.4 |

**Table 2.** *Cont.*

| Soil Parameter | Calcisols (N = 16) | | | |
|---|---|---|---|---|
| | **Min** | **Max** | **Mean ± SD** | **CV (%)** |
| | Regosols (N = 12) | | | |
| Clay (%) | 20.0 | 28.5 | 24.3 ± 3.2 | 12.1 |
| Silt (%) | 31.5 | 40.0 | 36.5 ± 2.9 | 8.1 |
| Sand(%) | 33.0 | 47.5 | 39.3 ± 4.6 | 11.8 |
| CEC (cmol kg$^{-1}$) | 14.2 | 25.0 | 18.8 ± 3.9 | 21.1 |
| pH | 7.4 | 7.9 | 7.7 ± 0.2 | 2.0 |
| EC (dS m$^{-1}$) | 0.7 | 1.0 | 0.8 ± 0.2 | 19.5 |
| OM (%) | 0.9 | 9.6 | 3.2 ± 2.3 | 105.2 |
| CCE (%) | 8.5 | 44.5 | 25.1 ± 12.4 | 49.3 |

CEC—Cation Exchangeable Capacity; EC—Electrical Conductivity; OM—Organic Matter; CCE—Calcium Carbonate Equivalent; SD—Standard Deviation; CV—Coefficient of Variation.

### 3.2. Soil Heavy Metals

Table 3 shows the range, mean, standard deviation, and coefficient of variations for the studied elements in different soil types. The concentration of the heavy metals in the soil samples varied in the range of 29.9–284.9 mg kg$^{-1}$ for Zn, 10.1–221.4 mg kg$^{-1}$ for Cu, 0.49–0.91 mg kg$^{-1}$ for Cd, 21.9–99.6 mg kg$^{-1}$ for Pb, and 10.3–122.1 mg kg$^{-1}$ for Ni.

**Table 3.** Summary statistics of heavy metal concentrations of the soils.

| Heavy Metal | Calcisols (N = 16) | | | |
|---|---|---|---|---|
| | **Min** | **Max** | **Mean ± SD** | **CV (%)** |
| Zn (mg kg$^{-1}$) | 59.8 | 118.3 | 77.7 [c] ± 17.1 | 22.0 |
| Cu (mg kg$^{-1}$) | 18.28 | 50.38 | 34.45 [b] ± 8.43 | 14.5 |
| Cd (mg kg$^{-1}$) | 0.54 | 0.98 | 0.76 [a] ± 0.13 | 47.1 |
| Pb (mg kg$^{-1}$) | 24.28 | 99.55 | 51.69 [a] ± 22.92 | 52.3 |
| Ni (mg kg$^{-1}$) | 12.58 | 120.38 | 61.16 [a] ± 41.99 | 18.6 |
| | Cambisols (10) | | | |
| Zn (mg kg$^{-1}$) | 75.6 | 110.8 | 87.8 [b] ± 15.0 | 17.1 |
| Cu (mg kg$^{-1}$) | 13.1 | 56.3 | 38.7 [b] ± 17.1 | 14.2 |
| Cd (mg kg$^{-1}$) | 0.5 | 0.9 | 0.7 [a] ± 0.2 | 44.5 |
| Pb (mg kg$^{-1}$) | 25.9 | 66.6 | 43.0 [c] ± 18.6 | 43.3 |
| Ni (mg kg$^{-1}$) | 15.4 | 113.1 | 41.9 [b] ± 22.0 | 22.5 |
| | Fluvisols (12) | | | |
| Zn (mg kg$^{-1}$) | 57.4 | 284.9 | 96.6 [a] ± 60.4 | 36.5 |
| Cu (mg kg$^{-1}$) | 26.8 | 221.4 | 61.1 [a] ± 41.9 | 18.5 |
| Cd (mg kg$^{-1}$) | 0.6 | 0.9 | 0.8 [a] ± 0.1 | 68.2 |
| Pb (mg kg$^{-1}$) | 30.7 | 88.0 | 53.8 [a] ± 22.5 | 69.7 |
| Ni (mg kg$^{-1}$) | 13.3 | 100.4 | 45.4 [b] ± 31.6 | 12.5 |
| | Regosols (N = 12) | | | |
| Zn (mg kg$^{-1}$) | 56.7 | 125.9 | 84.4 [b] ± 22.8 | 27.0 |
| Cu (mg kg$^{-1}$) | 10.1 | 40.0 | 28.8 [c] ± 8.5 | 19.5 |
| Cd (mg kg$^{-1}$) | 0.5 | 0.8 | 0.7 [a] ± 0.1 | 41.4 |
| Pb (mg kg$^{-1}$) | 21.9 | 77.7 | 48.7 [b] ± 19.2 | 49.4 |
| Ni (mg kg$^{-1}$) | 10.3 | 122.1 | 38.3 [c] ± 35.6 | 13.2 |

SD—Standard deviation; CV—Coefficient of variation. Values with the same lowercase letters within the means column are not significantly different at $p < 0.05$.

The average values of these elements in different soil types were in the order of Flu > Cam > Reg > Cal for Zn, Flu > Cam > Cal > Reg for Cu and Pb, and Cal > Flu > Cam > Reg for Cd and Ni. These sequences show that the highest and lowest mean values

of these elements occurred in Fluvisols and Regosols, respectively, for most elements. Previous studies have also reported significant differences in heavy metals among different soil types [7,31]. The background values, measured using the median absolute deviation method [26,32], were 53 mg kg$^{-1}$ for Zn, 32.4 mg kg$^{-1}$ for Cu, 0.42 mg kg$^{-1}$ for Cd, 39.3 mg kg$^{-1}$ for Pb, and 36.1 mg kg$^{-1}$ for Ni. These are almost similar to those values reported for soils in other parts of the world [33]. There is a consensus in the world that the comparison of element concentrations in urban soils with the values measured in other cities is a good way to estimate the pollution of these soils and apply sound management practices [2].

### 3.3. Coefficient of Correlation

Coefficients of correlation between the heavy metals and the physicochemical characteristics were calculated and the obtained results are reported in Table 4. Organic matter showed a significant positive correlation with Cu and Zn, implying the high tendency of these elements to form complexes with organic compounds. Shaheen et al. [34] and Zhou et al. [35] reported the impact and high capability of organic matter in maintaining, fixing, and distributing Cu and Zn in Germany and China, respectively. Although past studies have revealed a high correlation of clay distribution and CEC with heavy metals [6,34,36], we did not observe this behavior in the present investigation. This may be associated with the fact that the soils in our study site are alkaline, calcareous, and rich in base cations (Ca, Mg, Na, and K) and these cations fill exchangeable sites fast and do not allow the absorption of heavy metals. This has been well explained by Rezapour and Moazzeni [37]. Another possible explanation could be attributed to the soil's mineralogical constituents, where illite, chlorite, and kaolinite together constitute more than 80% of the mineral constituents in the clay fraction of these soils [37]. Due to a low CEC and surface area, clay minerals such as illite and chlorite have a lower affinity to absorb heavy metals, resulting in the weak influence of the clay fraction on the adsorption and retention of heavy metals [33,37]. Ni ($p < 0.01$), Zn ($p < 0.05$), and Cu ($p < 0.05$) showed a significant positive correlation with $Al_2O_3$ and $Fe_2O_3$, which may imply a strong relationship and controlling role of aluminosilicate in distributing these elements. The important role of aluminum and iron oxides in maintaining and absorbing heavy metals has already been proven [33]. Soil EC and pH were insignificantly correlated with heavy metals. In semi-arid regions like our study site, soil EC variations are controlled by elements that are more soluble and abundant than heavy metals (e.g., Na and K), so the insignificant correlation of EC and heavy metals in this study seems reasonable. The lack of a correlation between soil pH and heavy metals can also be attributed to the narrow range of pH in these soils (7.36–8.06), as its small coefficient of variations (CV = 2.2%) confirms too. Also, significant correlations were observed between Cd and Pb and between Ni and Cu, which may reflect that their origins and distributions are controlled by the same processes. Similar correlations have been reported for different soils throughout the world in several authentic research works, e.g., Chabukdhara and Nema [38] in India, Liu et al. [39] in China, and Rinklebe et al. [36] in Germany. All in all, the concentrations of heavy metals in this study did not exhibit similar relationships with the soil characteristics and this can arise from differences in the anthropogenic and lithogenic sources of these elements.

**Table 4.** Correlation coefficients of heavy metals and selected soil properties in the soils (* $p < 0.05$; ** $p < 0.01$) (N = 50).

| | Zn | Cu | Cd | Pb | Ni | $Al_2O_3$ | $Fe_2O_3$ | Clay | Silt | Sand | pH | EC | OC | CEC |
|---|---|---|---|---|---|---|---|---|---|---|---|---|---|---|
| Zn | 1.00 | | | | | | | | | | | | | |
| Cu | 0.26 | 1.00 | | | | | | | | | | | | |
| Cd | 0.47 ** | 0.08 | 1.00 | | | | | | | | | | | |
| Pb | 0.29 | 0.05 | 0.61 ** | 1.00 | | | | | | | | | | |
| Ni | 0.23 | 0.59 ** | −0.26 | −0.22 | 1.00 | | | | | | | | | |
| $Al_2O_3$ | 0.39 * | 0.34 * | 0.23 | 0.27 | 0.58 ** | 1.00 | | | | | | | | |
| $Fe_2O$ | 0.31 * | 0.36 * | 0.21 | 0.20 | 0.55 ** | 0.89 ** | 1.00 | | | | | | | |
| Clay | 0.15 | 0.07 | 0.01 | 0.02 | 0.09 | 0.47 * | 0.43 * | 1.00 | | | | | | |
| Silt | 0.05 | −0.04 | 0.02 | −0.04 | 0.05 | 0.06 | 0.03 | 0.29 * | 1.00 | | | | | |
| Sand | −0.14 | −0.06 | −0.03 | −0.04 | −0.11 | −0.29 | −0.24 | −0.23 | −0.07 | 1.00 | | | | |
| pH | −0.05 | −0.02 | 0.00 | 0.00 | −0.03 | −0.20 | −0.15 | 0.17 | 0.00 | −0.12 | 100 | | | |
| EC | 0.08 | −0.05 | −0.09 | 0.00 | −0.01 | −0.02 | −0.03 | 0.01 | 0.06 | 0.09 | −0.31 * | 1.00 | | |
| OC | 0.32 * | 0.53 ** | 0.19 | 0.16 | 0.21 | 0.18 | 0.16 | 0.28 | 0.02 | −0.05 | −0.20 | 0.24 | 1.00 | |
| CEC | 0.15 | 0.21 | 0.04 | −0.01 | 0.00 | 0.15 | 0.13 | 0.41 * | 0.10 | −0.07 | −0.27 | 0.00 | 0.36 * | 1.00 |

*3.4. Factor Analysis*

The results of the principal component analysis (PCA), which is an effective way to determine the origin of elements, are given in Table 5. The results of the KMO test (0.635) and Bartlett's test ($p < 0.000$) supported that the data were suitable for PCA. In each principal component (PC), the factor loading of the elements can be in the range of strong (>0.75), moderate (0.5–0.75), and weak (0.3–0.5) [40].

**Table 5.** Results of the principal component analysis (PCA) for heavy metals in the soils.

| Principal Components | PC1 | PC2 | PC3 |
|---|---|---|---|
| Eigenvalues | 2.020 | 1.804 | 1.683 |
| Variance (%) | 31.612 | 28.18 | 14.684 |
| Cumulative variance (%) | 31.60 | 59.781 | 74.531 |
| Zn | 0.14 | 0.89 | 0.078 |
| Cu | −0.08 | 0.11 | 0.87 |
| Cd | 0.76 | 0.16 | −0.09 |
| Pb | 0.81 | −0.17 | −0.07 |
| Ni | −0.04 | −0.21 | 0.76 |
| Clay | −0.19 | −0.18 | 0.23 |
| CEC | −0.13 | 0.19 | 0.21 |
| OM | −0.07 | 0.14 | 0.51 |
| CCE | 0.37 | 0.21 | −0.08 |

According to Table 5, heavy metals were grouped into three principal components and accounted for 74% of the total variance. The first principal component (PC1), which captured 32.6% of the total variance, was the most important in determining the variations in the heavy metal concentrations, and the next two PC accounted for 28.2 and 14.75% of the total variance, respectively. PC1 contained Pb and Cd with a strong factor load (PC > 0.75) and a significant positive correlation was established between them (Table 4). This may reflect a similar pollution level and common pollution sources for these two metals [41–43]. Pb and Cd concentrations were higher in most soil samples than their background values and they also had a relatively high coefficient of variations (CV > 35%) in different soil types (Table 3). This means that these elements have mainly anthropogenic origins. The main role of anthropogenic origins in the accumulation of Cd and Pb in urban soils has been stressed in several other studies around the world [15,42–44]. In the study site, a huge amount of urban waste containing industrial human wastes spread into the soils via different factors (e.g., wind, water, humans). Based on the literature, these materials contain significant amounts of Pb, Cd, and Zn [6,45–47]. However, the role of traffic and agrochemicals in the accumulation of these elements should not be overlooked. The second component (PC2) is more complicated than the other two components. Its main element is Zn with a strong factor loading (0.89), standing between the human factor (PC1) and the natural factor (PC3). So, Zn is likely to have a common human–natural origin. The concentration of Zn was higher than its background value in 61% of the soil samples while in 39% of the samples, it was lower than the background value. Its coefficient of variations was in a moderate range (15% < CV < 35%) within different soil types. These variations may confirm the concurrent anthropogenic and natural origins of Zn. Using multivariate statistical and isothermic methods, Lv and Liu [48] and Hu et al. [10] reported such a human–natural origin for Zn in peri-urban agricultural soils, though the human side of the origins was stronger. Ni and Cu concentrations were smaller than or almost equal to their background concentrations in most soil samples and they had a low coefficient of variations (CV < 15%). This implies that the third component (PC3), whose main factor loadings come from Ni (0.76) and Cu (0.87), has a natural origin and the concentration of the elements of this group is mainly controlled by the weathering of parent material and pedogenic processes. The significant impact of natural factors on the concentration of Ni and Cu has been reported by other researchers such as Franco-Uría et al. [49] in Spain, Adimalla [15] in India, and Yang et al. [50] in China.

### 3.5. Soil Pollution Indices

Soil pollution indices including Pi, PIN, and PLI were calculated using Equations (1)–(3), respectively. The results are summarized in Tables 6 and 7. The PI value for Zn, Cu, Cd, Pb, and Ni was in the ranges of 0.9–8.8, 0.1–6.32, 1.26–2.16, 0.71–3.21, and 0.31–7.3, respectively. The data show that, in terms of Pi, the studied soils were in the range of unpolluted (PI $\leq$ 1) to extremely polluted (PI > 6). Overall, 100%, 78.5%, and 82.5% of the soil samples were lowly (1 < PI $\leq$ 2) to moderately (2 < PI $\leq$ 3) polluted by Cd, Pb, and Zn, respectively. Most soil samples were in the class of clean soils in terms of Cu and Ni, implying that most of them are not polluted by these two elements.

**Table 6.** Percentages of class distribution for PI in total soil samples.

| Pollution Class | Zn | Cu | Cd | Pb | Ni |
|---|---|---|---|---|---|
| Clean | - | 70 | - | 20 | 47.5 |
| Low | 32.5 | 25 | 80 | 45 | 22.5 |
| Moderate | 50 | - | 20 | 32.5 | 10 |
| High | 12.5 | - | - | 2.5 | 20 |
| Very High | 5 | 5 | - | - | - |

Clean: PI $\leq$ 1; Low: 1 < PI $\leq$ 2; Moderate: 2 < PI $\leq$ 3; High: 3 < PI $\leq$ 6; Very High: PI > 6.

**Table 7.** Percentage of class distribution for PIN in different soil types.

| Soil Type | Min | Max | Mean | PIN Classes (% of Total Sample) | | |
|---|---|---|---|---|---|---|
| | | | | L | M | H |
| Calcisols | 1.81 | 3.28 | 2.66 | 12.5 | 50 | 37.5 |
| Cambisols | 2.21 | 3.12 | 2.59 | ------ | 75 | 25 |
| Fluvisols | 1.87 | 6.88 | 3.1 | 16.7 | 58.3 | 25 |
| Regosols | 1.8 | 3.31 | 2.49 | 12.5 | 62.5 | 25 |

L: Low (1 < PIN < 2), M: Moderate (2 < PIN < 3), H: High (PIN > 3).

The PIN index varied from 1.8 to 6.9 with an average of 2.7 in different soil types, reflecting a range of lowly polluted (1 < PIN < 2) to highly polluted (PIN > 3) classes. In terms of the mean PIN value, the soil types were in the order of Flu (3.1) > Cal (2.66) > Cam (2.59) > Reg (2.47). This means that different soil types had not been influenced by anthropogenic processes equally. In a study on the heavy metal pollution potential of soils in central Greece with different soil types (Alfisols, Inceptisols, Vertisols, and Entisols), Golia et al. [31] concluded that the soil type played a significant role in the quantity and class of pollution indices.

PLI was in the range of 0.35–3.9 with a mean of 1.42. The average variations of this index for different soil types were in the order of Flu (1.53) > Cal (1.49) > Cam (1.36) > Reg (1.29) (Figure 2), which is identical to the order observed for PIN. Zn and Cd were found to be the most influential on PLI. The average of this index, examined for the four soil types, showed a moderate pollution class (1 < PLI < 2). In total, as far as the PIN and PLI indices are concerned, most soils in the study site are in the moderately polluted class, which corroborates other studies in other parts of the world [15,43].

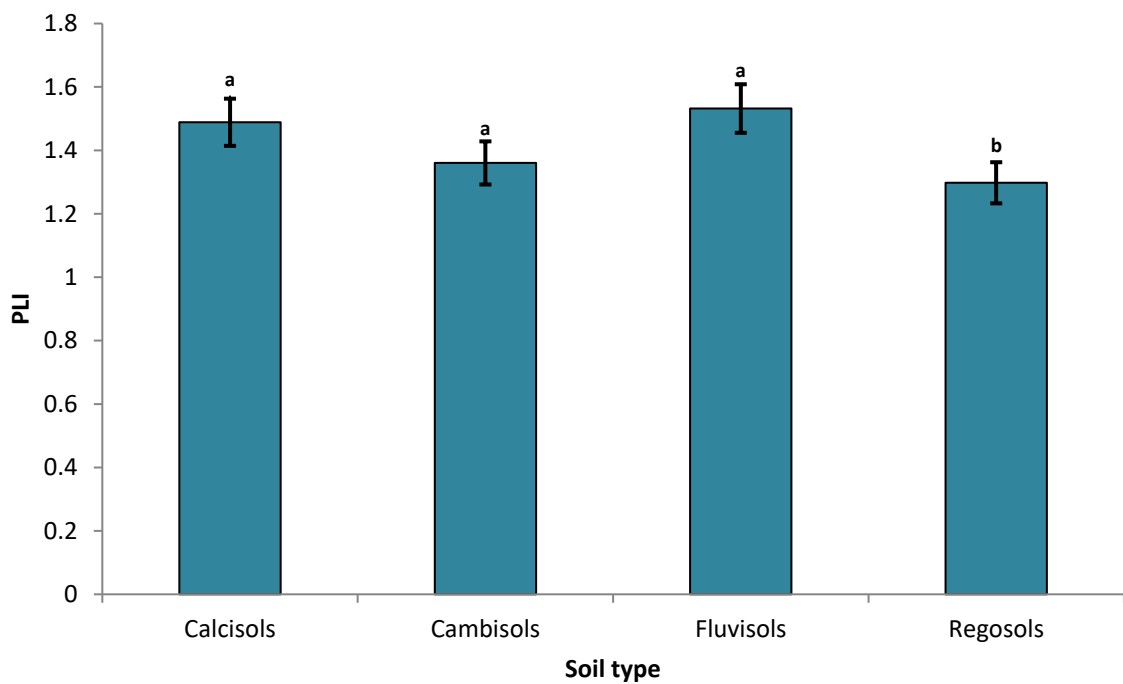

**Figure 2.** Comparison of the mean values of the pollution load index (PLI) for different soil types. Different letters represent significant differences in the PLI values between soil types at the $p < 0.05$ confidence level using Duncan's test.

*3.6. Health Risk Assessment*

The health risk of the five elements to induce cancerous and non-cancerous diseases via ingestion, dermal contact, and inhalation was calculated for children, adult men, and adult women. Table 8 shows the minimum, maximum, and average non-carcinogenic risk for individual heavy metals taken through ingestion, inhalation, and dermal contact in different soil types and for children, adult women, and adult men separately.

As is evident, $HQ_{ing}$ has the maximum value for all elements for all age groups. $HQ_{der}$ and $HQ_{inh}$ are in the next ranks. This means that the non-carcinogenic risk of diseases caused by the ingestion of heavy metals in the soils of the region is much more dangerous than that by dermal contact and inhalation. The higher risk of heavy metals taken by ingestion versus the other absorption pathways has been stated in other research too [15,43]. In all soil types, children had the highest HQ via ingestion, inhalation, and dermal contact, and adult females and males were in the next ranks. This means that children are exposed to the risk of non-cancerous diseases by heavy metals to a much greater extent than adults. However, in all soil samples, HQ was <1 in all three exposure pathways of ingestion, inhalation, and dermal contact. So, the risk of non-cancerous alimentary, respiratory, and dermal diseases is weak for all three studied population groups [28]. In addition, in most soil samples and among the three exposure pathways, HQ for the three population groups was the highest for Pb followed by HQ-Cd, HQ-Zn, HQ-Ni, and HQ-Cu. This finding implies a higher risk of the involvement of Pb in the incidence of non-cancerous diseases than the other elements, which is consistent with previous studies [51,52]. For all age groups, the changes in the three exposure pathways (ingestion, inhalation, and dermal contact) for the studied heavy metals in different soil types were in the order of Flu > Cal > Reg > Cam. This sequence shows that the residents of Fluvisols are more exposed to the health risk of heavy metals to induce non-cancerous diseases than the residents of the other soil types.

Table 8. Exposure values of non-carcinogenic risks (HI) for different population groups (N = 50).

| Heavy Metal | | Child | | | Adults-Male | | | Adults-Female | | |
|---|---|---|---|---|---|---|---|---|---|---|
| | | Ingestion | Inhalation | Dermal | Ingestion | Inhalation | Dermal | Ingestion | Inhalation | Dermal |
| Zn | Min | $2.45 \times 10^{-3}$ | $6.75 \times 10^{-8}$ | $3.2624 \times 10^{-5}$ | $1.91 \times 10^{-4}$ | $2.79 \times 10^{-8}$ | $2.62 \times 10^{-6}$ | $2.23 \times 10^{-5}$ | $3.29 \times 10^{-8}$ | $3.08 \times 10^{-6}$ |
| | Max | $5.38 \times 10^{-2}$ | $3.39 \times 10^{-7}$ | $1.63 \times 10^{-4}$ | $9.59 \times 10^{-4}$ | $1.40 \times 10^{-7}$ | $1.31 \times 10^{-5}$ | $1.12 \times 10^{-3}$ | $1.65 \times 10^{-7}$ | $1.54 \times 10^{-5}$ |
| | Mean | $5.22 \times 10^{-3}$ | $1.02 \times 10^{-7}$ | $4.92 \times 10^{-5}$ | $2.89 \times 10^{-4}$ | $4.22 \times 10^{-8}$ | $3.97 \times 10^{-6}$ | $2.77 \times 10^{-4}$ | $4.97 \times 10^{-8}$ | $4.65 \times 10^{-6}$ |
| | SD | $8.20 \times 10^{-3}$ | $4.38 \times 10^{-8}$ | $2.11 \times 10^{-5}$ | $1.24 \times 10^{-4}$ | $1.81 \times 10^{-8}$ | $1.70 \times 10^{-6}$ | $1.86 \times 10^{-4}$ | $2.13 \times 10^{-8}$ | $2.00 \times 10^{-6}$ |
| Cu | Min | $5.93 \times 10^{-3}$ | $1.62 \times 10^{-7}$ | $5.25 \times 10^{-5}$ | $4.61 \times 10^{-4}$ | $6.72 \times 10^{-8}$ | $4.23 \times 10^{-6}$ | $5.39 \times 10^{-4}$ | $7.91 \times 10^{-8}$ | $4.96 \times 10^{-6}$ |
| | Max | $1.79 \times 10^{-1}$ | $1.96 \times 10^{-6}$ | $6.36 \times 10^{-4}$ | $5.58 \times 10^{-3}$ | $8.14 \times 10^{-7}$ | $5.12 \times 10^{-5}$ | $6.53 \times 10^{-3}$ | $9.58 \times 10^{-7}$ | $6.01 \times 10^{-5}$ |
| | Mean | $1.91 \times 10^{-2}$ | $3.77 \times 10^{-7}$ | $1.22 \times 10^{-4}$ | $1.07 \times 10^{-3}$ | $1.56 \times 10^{-7}$ | $9.84 \times 10^{-6}$ | $1.25 \times 10^{-3}$ | $1.83 \times 10^{-7}$ | $1.15 \times 10^{-5}$ |
| | SD | $2.98 \times 10^{-2}$ | $3.67 \times 10^{-7}$ | $1.19 \times 10^{-4}$ | $1.04 \times 10^{-3}$ | $1.52 \times 10^{-7}$ | $9.57 \times 10^{-6}$ | $1.22 \times 10^{-3}$ | $1.78 \times 10^{-7}$ | $1.12 \times 10^{-5}$ |
| Cd | Min | $6.89 \times 10^{-3}$ | $1.91 \times 10^{-7}$ | $3.70 \times 10^{-4}$ | $5.35 \times 10^{-4}$ | $7.84 \times 10^{-8}$ | $2.94 \times 10^{-5}$ | $6.25 \times 10^{-4}$ | $9.22 \times 10^{-8}$ | $3.45 \times 10^{-5}$ |
| | Max | $1.58 \times 10^{-1}$ | $3.49 \times 10^{-7}$ | $6.76 \times 10^{-4}$ | $9.9 \times 10^{-4}$ | $1.45 \times 10^{-7}$ | $5.44 \times 10^{-5}$ | $1.15 \times 10^{-3}$ | $1.70 \times 10^{-7}$ | $6.38 \times 10^{-5}$ |
| | Mean | $1.38 \times 10^{-2}$ | $2.72 \times 10^{-7}$ | $5.31 \times 10^{-4}$ | $7.33 \times 10^{-4}$ | $1.07 \times 10^{-7}$ | $4.03 \times 10^{-5}$ | $8.57 \times 10^{-4}$ | $1.26 \times 10^{-7}$ | $4.73 \times 10^{-5}$ |
| | SD | $2.37 \times 10^{-2}$ | $4.71 \times 10^{-8}$ | $9.11 \times 10^{-5}$ | $1.33 \times 10^{-4}$ | $1.95 \times 10^{-8}$ | $7.33 \times 10^{-6}$ | $1.56 \times 10^{-4}$ | $2.29 \times 10^{-8}$ | $8.60 \times 10^{-6}$ |
| Pb | Min | $9.01 \times 10^{-2}$ | $2.46 \times 10^{-6}$ | $1.58 \times 10^{-3}$ | $7.47 \times 10^{-4}$ | $1.02 \times 10^{-6}$ | $1.27 \times 10^{-4}$ | $8.18 \times 10^{-3}$ | $1.19 \times 10^{-6}$ | $1.49 \times 10^{-4}$ |
| | Max | 3.07 | $1.00 \times 10^{-5}$ | $6.48 \times 10^{-3}$ | $2.87 \times 10^{-2}$ | $4.20 \times 10^{-6}$ | $5.22 \times 10^{-4}$ | $3.35 \times 10^{-2}$ | $4.92 \times 10^{-6}$ | $6.12 \times 10^{-4}$ |
| | Mean | $2.74 \times 10^{-2}$ | $5.24 \times 10^{-6}$ | $3.41 \times 10^{-3}$ | $1.36 \times 10^{-2}$ | $2.15 \times 10^{-6}$ | $2.68 \times 10^{-4}$ | $1.72 \times 10^{-2}$ | $2.52 \times 10^{-6}$ | $3.14 \times 10^{-4}$ |
| | SD | $4.62 \times 10^{-1}$ | $2.15 \times 10^{-6}$ | $1.77 \times 10^{-3}$ | $7.19 \times 10^{-3}$ | $8.95 \times 10^{-7}$ | $1.11 \times 10^{-4}$ | $7.16 \times 10^{-3}$ | $1.05 \times 10^{-6}$ | $1.31 \times 10^{-4}$ |
| Ni | Min | $8.65 \times 10^{3}$ | $2.46 \times 10^{-7}$ | $9.08 \times 10^{-5}$ | $6.72 \times 10^{-4}$ | $9.56 \times 10^{-8}$ | $6.85 \times 10^{-6}$ | $7.85 \times 10^{-4}$ | $1.12 \times 10^{-7}$ | $8.03 \times 10^{-6}$ |
| | Max | $6.75 \times 10^{-1}$ | $2.08 \times 10^{-6}$ | $7.69 \times 10^{-4}$ | $6.16 \times 10^{-3}$ | $8.77 \times 10^{-7}$ | $6.28718 \times 10^{-5}$ | $7.20 \times 10^{-3}$ | $1.03 \times 10^{-6}$ | $7.37 \times 10^{-5}$ |
| | Mean | $5.38 \times 10^{-2}$ | $1.12 \times 10^{-6}$ | $4.04 \times 10^{-4}$ | $2.73 \times 10^{-3}$ | $3.89 \times 10^{-7}$ | $2.79 \times 10^{-5}$ | $3.19 \times 10^{-3}$ | $4.57 \times 10^{-7}$ | $3.27 \times 10^{-5}$ |
| | SD | $1.03 \times 10^{-1}$ | $6.24 \times 10^{-7}$ | $2.3 \times 10^{-4}$ | $1.82 \times 10^{-3}$ | $2.59 \times 10^{-7}$ | $1.85 \times 10^{-5}$ | $2.12 \times 10^{-3}$ | $3.04 \times 10^{-7}$ | $2.17 \times 10^{-5}$ |

The HI value, which is the sum of HQ of the five elements in each exposure way, was <1 for all three exposure pathways (ingestion, inhalation, and dermal contact) in all soil types and all three groups of children, women, and men, implying that these elements have a weak non-carcinogenic risk, even altogether (Table 8). For all soil types, the HI of ingestion was significantly higher than that of the dermal contact and inhalation, and dermal contact exhibited a significantly higher HI than inhalation. Similar results have been reported by Chen et al. [53] and Ma et al. [54] for the soils in China. We also found that the HI of the three exposure pathways in different age groups was in the order of children > adult females > adult males, which is similar to the sequence observed for HQ. As for the ingestion and inhalation of heavy metals, Pb and Ni had the highest impact on the HI quantity followed by Cd, Cu, and Zn. But the HI of dermal contact was most affected by Pb and Cd, followed by Ni, Cu, and Zn. Similar results were revealed for all soil types and all age groups and are in agreement with previous studies [55]. The differences found in the impact of the elements on HI may be related to their different toxicity mechanisms and/or the difference in reference dose (RfD) of these elements [6].

THI, which was the sum of the HI in the three exposure ways of the heavy metals, was in the order of children > adult females > adult males (Figure 3). This is in agreement with the research on urban soils in other parts of the world [9,38]. The THI value of children was several times as great as that of adults, as described below. As for children, THI-Calcisols was 12–13 times as great as for adult males and 10.7–11 times as great as for adult females; THI-Cambisols was 32–50 times as great as for adult males and 11–11.4 times as great as for adult females; THI-Fluvisols was 12–13 times as great as for adult males and 11–11.2 times as great as for adult females; and THI-Regosols was 12.6–12.7 times as great as for adult males and 11–12 times as great as for adult females This implies that the risk of non-cancerous diseases varies significantly among different soil types. The variations in THI among different age types were in the range of 0.2–0.4 for children, 0.003–0.004 for adult males, and 0.04–0.1 for adult females. These results reflect that, just like HI, THI was <1 for all soil samples and all population groups, so the local residents are not much threatened by the risk of non-cancerous diseases arising from the total of the three exposure pathways and the elements [28]. Overall, considering HQ, HI, and THI, it becomes evident that children are more exposed to the risk of non-cancerous diseases by these elements when compared to adult males and females. The reasons can be sought in children's behavioral and metabolic features such as (i) their higher rate of breathing per unit of body weight versus adults, (ii) their more outdoor activities, (iii) contact with different things, soil, and continuously sucking their hands and fingers, and (iv) their higher rate of digestive absorption of some elements versus adults [39]. Furthermore, in all three population groups, the highest HI and THI were related to Fluvisols, followed by Calcisols, Regosols, and Cambisols (Figure 3). In other words, the residents of Fluvisols are more exposed to the risk of non-cancerous diseases induced by heavy metals than the residents of other soil types, which was observed with HQ too.

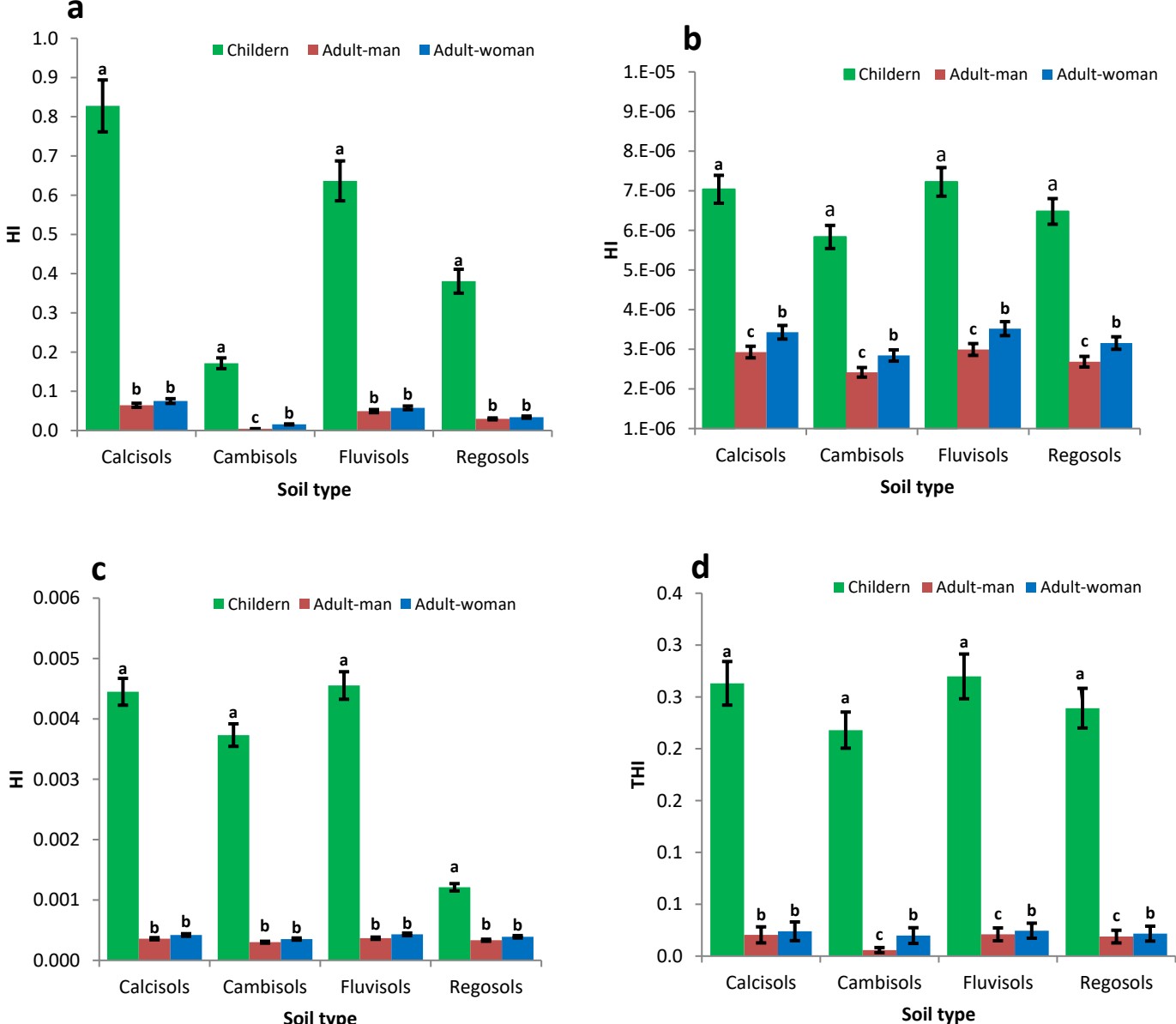

**Figure 3.** Comparison of the mean values of hazard index (HI) for pathways of ingestion (**a**), inhalation (**b**), dermal contact (**c**), and total exposure hazard index (THI, (**d**)) in different soil types. Different letters represent significant differences in the HI and THI values between soil types at a $p < 0.05$ confidence level using Duncan's test.

### 3.7. Cancergenic Risk

The risk of cancerous diseases was assessed for the three population groups in terms of the carcinogenic risk (CR) and total carcinogenic risk (TCR) for Cd, Pb, and Ni which people were exposed to through ingestion ($CR_{ing}$), inhalation ($CR_{inh}$), and dermal contact ($CR_{der}$) (Table 9). It was found that CR of these elements was in the order of Cd > Ni > Pb across all the exposure pathways and population groups. Similarly, Qing et al. [56] and Ma et al. [54] reported that Cd and Ni had the highest CR values for children and adults.

**Table 9.** Exposure values of carcinogenic risks (CR) for different population groups (N = 50).

| Heavy Metals | | Child | | | Adults-Male | | | Adult-Female | | |
|---|---|---|---|---|---|---|---|---|---|---|
| | | Ingestion | Inhalation | Dermal | Ingestion | Inhalation | Dermal | Ingestion | Inhalation | Dermal |
| Ni | Mean | $5.14 \times 10^{-4}$ | $5.61 \times 10^{-9}$ | $1.23 \times 10^{-6}$ | $3.02 \times 10^{-5}$ | $5.31 \times 10^{-9}$ | $8.33 \times 10^{-8}$ | $5.3 \times 10^{-5}$ | $7.81 \times 10^{-9}$ | $1.46 \times 10^{-7}$ |
| | SD | $4.02 \times 10^{-4}$ | $3.99 \times 10^{-9}$ | $1.07 \times 10^{-6}$ | $3.11 \times 10^{-5}$ | $4.61 \times 10^{-9}$ | $8.58 \times 10^{-8}$ | $3.48 \times 10^{-5}$ | $5.17 \times 10^{-9}$ | $9.68 \times 10^{-8}$ |
| | Max | $1.28 \times 10{-3}$ | $1.61 \times 10^{-8}$ | $3.40 \times 10^{-6}$ | $9.96 \times 10^{-5}$ | $1.51 \times 10^{-8}$ | $2.74 \times 10^{-7}$ | $1.21 \times 10^{-4}$ | $1.78 \times 10^{-8}$ | $3.34 \times 10^{-7}$ |
| | Min | $1.03 \times 10^{-4}$ | $1.42 \times 10^{-9}$ | $4.11 \times 10^{-8}$ | $8.02 \times 10^{-6}$ | $1.23 \times 10^{-9}$ | $2.21 \times 10^{-8}$ | $1.32 \times 10^{-5}$ | $1.94 \times 10^{-9}$ | $3.64 \times 10^{-8}$ |
| Pb | Mean | $1.85 \times 10^{-4}$ | $4.88 \times 10^{-9}$ | $4.56 \times 10^{-7}$ | $1.44 \times 10^{-5}$ | $2.11 \times 10^{-9}$ | $5.47 \times 10^{-8}$ | $1.68 \times 10^{-5}$ | $2.48 \times 10^{-9}$ | $4.66 \times 10^{-8}$ |
| | SD | $7.74 \times 10^{-5}$ | $2.25 \times 10^{-9}$ | $2.41 \times 10^{-7}$ | $6.009 \times 10^{-6}$ | $8.80 \times 10^{-10}$ | $3.77 \times 10^{-8}$ | $6.97 \times 10^{-6}$ | $1.03 \times 10^{-9}$ | $1.93 \times 10^{-8}$ |
| | Max | $3.62 \times 10^{-4}$ | $9.95 \times 10^{-9}$ | $9.61 \times 10^{-7}$ | $2.81 \times 10^{-5}$ | $4.12 \times 10^{-9}$ | $1.77 \times 10^{-7}$ | $3.28 \times 10^{-5}$ | $4.85 \times 10^{-9}$ | $9.08 \times 10^{-8}$ |
| | Min | $8.84 \times 10^{-5}$ | $1.26 \times 10^{-9}$ | $2.36 \times 10^{-8}$ | $6.86 \times 10^{-6}$ | $1.005 \times 10^{-9}$ | $1.88 \times 10^{-8}$ | $8.02 \times 10^{-6}$ | $1.18 \times 10^{-9}$ | $2.21 \times 10^{-8}$ |
| Cd | Mean | $1.41 \times 10^{-4}$ | $3.69 \times 10^{-9}$ | $5.41 \times 10^{-7}$ | $1.1 \times 10^{-5}$ | $1.61 \times 10^{-9}$ | $3.02 \times 10^{-8}$ | $1.28 \times 10^{-5}$ | $1.89 \times 10^{-9}$ | $3.5 \times 10^{-8}$ |
| | SD | $2.56 \times 10^{-5}$ | $9.15 \times 10^{-10}$ | $5.54 \times 10^{-7}$ | $1.99 \times 10^{-6}$ | $2.92 \times 10^{-10}$ | $5.50 \times 10^{-9}$ | $2.32 \times 10^{-6}$ | $3.44 \times 10^{-10}$ | $6.45 \times 10^{-9}$ |
| | Max | $1.91 \times 10^{-4}$ | $5.24 \times 10^{-9}$ | $3.45 \times 10^{-6}$ | $1.4847 \times 10^{-5}$ | $2.17 \times 10^{-9}$ | $4.08 \times 10^{-8}$ | $1.73 \times 10^{-5}$ | $2.55 \times 10^{-9}$ | $4.79 \times 10^{-8}$ |
| | Min | $1.03 \times 10^{-4}$ | $1.43 \times 10^{-9}$ | $2.68 \times 10^{-8}$ | $8.02 \times 10^{-6}$ | $1.17 \times 10^{-9}$ | $2.21 \times 10^{-8}$ | $9.38 \times 10^{-6}$ | $1.38 \times 10^{-9}$ | $2.59 \times 10^{-8}$ |

Based on the standards of the United States' Environmental Protection Agency [28], if the CR is smaller than $1 \times 10^{-6}$ (the likelihood of cancer incidence in one person per 1 million people), the carcinogenic risk is negligible, but CR > $1 \times 10^{-4}$ is unpermitted and dangerous to human health. A CR in the range of $1 \times 10^{-6}$ and $1 \times 10^{-4}$ shows the allowed risk which is under control. The CR of all three elements (Ni, Pb, and Cd) that the three population groups were exposed to via inhalation and dermal contact in all soil types was <$1 \times 10^{-6}$, so the risk of respiratory and dermal carcinogenicity of these metals among the local residents is negligible, but the result was different for ingestion. The $CR_{ing}$ for children was >$1 \times 10^{-4}$ in 100% of the soil samples for Cd, in 90% of the samples for Pb, and in 100% of the samples for Ni. As for adult males, it was >$1 \times 10^{-4}$ in 62.5% of the samples for Cd, in 5% of the samples for Pb, and in 25% of the samples for Ni. Concerning adult females, it was >$1 \times 10^{-4}$ in 87.5% of the samples for Cd, in 10% of the samples for Pb, and in 45% of the samples for Ni. These findings reflect that, just like the risk of non-cancerous diseases (HI) which turned out to be higher by ingestion (as already discussed), ingestion is again the most dangerous exposure pathway to cause cancers in the studied soils. The $CR_{ing}$ value among the population groups was in the order of children > adult females > adult males.

The high risk of cancers via ingestion in children in urban areas can be attributed to the fact that the roads and passages in these regions are mostly covered with dust and dirt, and children play with it and their fingers frequently come into contact with their mouths during playing. For all three elements of Cd, Pb, and Ni in all soil types and all three population groups, the CR of different exposure pathways were in the order of $CR_{ing} \gg CR_{der} > CR_{inh}$, which is similar to the risk of non-cancerous diseases (HQ and HI). This means that the risk of cancer by the ingestion of Cd, Pb, and Ni in the soils of the region is much higher than the risk of cancer by dermal contact or inhalation of these elements. These results fully agree with the reports of Pan et al. [43] and Yang et al. [50].

The TCR value in different soil types was >$1 \times 10^{-4}$ in children, but within the range of $1 \times 10^{-4}$ and $1 \times 10^{-6}$ for adult men and women (Figure 4). So, the total of ingestion, inhalation, and intentional/accidental contact with soil is in the impermissible range and hazardous to children, but it is within the controlled condition for adults. In all soil types and for all three elements, the TCR for the three population groups was most influenced by $CR_{ing}$ (99.2–99.6%), followed by $CR_{der}$ and $CR_{inh}$. The effect of $CR_{der}$ on the TCR was about 90–100 times as great as the effect of $CR_{inh}$. Similarly, Wu et al. [52] reported >90% effectiveness of $CR_{ing}$ in the TCR in the soils of north Tibet.

The highest values of soil pollution indices (PI, PIN, and PLI) and health risk indices (HQ, HI, THI, CR-Pb, CR-Cd, and TCR-Pb) were observed in Fluvisols, showing that this soil type is more polluted than the other types and has a higher potential to cause cancerous and non-cancerous risk. Fluvisols are naturally more fertile and have a higher potential for crop production [19,57]. Accordingly, these soils are more intensely cultivated (e.g., for summer crops, vegetables, corn, and sunflower) in the study site than the other types and they produce higher yields per unit area too. To gain these higher yields from these soils requires the application of various pesticides, herbicides, insecticides, and some chemical fertilizers whose formulations usually contain heavy metals and are extensively used in the region. Furthermore, Fluvisols in this region are located in the vicinity of an open wastewater canal that serves the utility systems of the urban area and farmers sometimes employ the water of this canal to irrigate their farms. So, these processes may be implicated in the significantly higher values of these indices in Fluvisols than in other soil types.

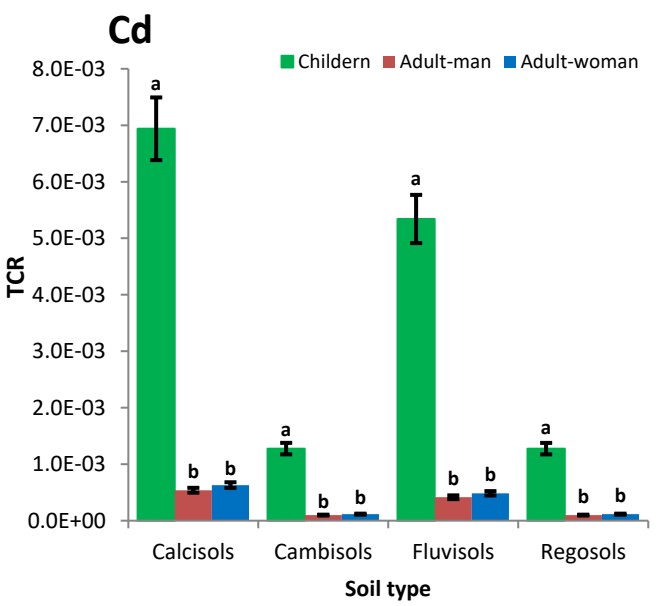

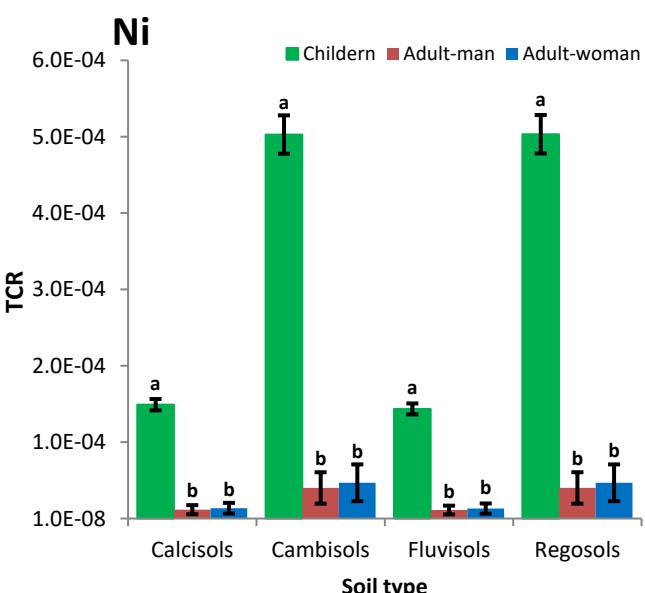

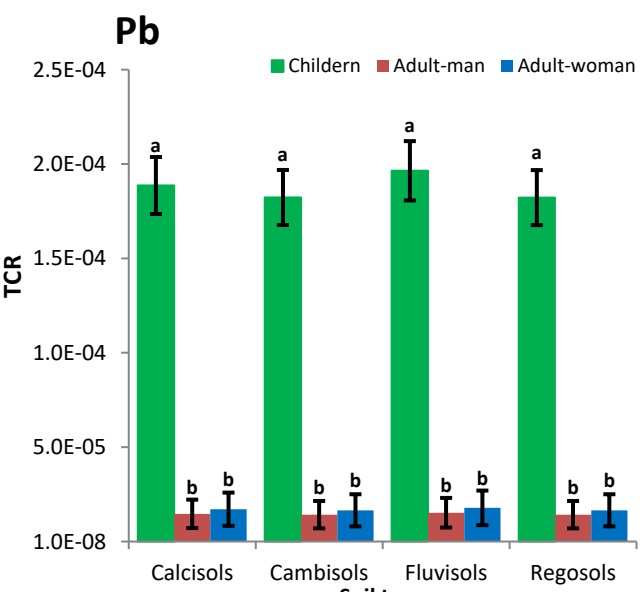

**Figure 4.** Comparison of the mean values of total carcinogenic risk (TCR) for the metals of Cd, Ni, and Pb in different soil types. Different letters represent significant differences in the TCR values between soil types at a $p < 0.05$ confidence level Duncan's test.

## 4. Conclusions

The soils in the study site included four types—Calcisols, Cambisols, Fluvisols, and Regosols. The average concentration of the studied five elements was in the order of Zn > Pb > Ni > Cu > Cd in all soil types. The highest average of most of these elements was observed in Fluvisols, followed by Calcisols, Cambisols, and Regosols. Based on PI, PIN, and PLI, most studied soils were found to be moderately polluted. In most soil samples, the non-carcinogenicity risk of all exposure pathways (THI) was the highest for children, followed by adult females and adult males. This implies that the non-carcinogenicity potential of the metals in all exposure pathways is more challenging for children. The carcinogenicity risk of Cd, Pb, and Ni for ingestion (CR) and for the total of exposure pathways (TCR) was estimated to be >1 × $10^{-4}$ for children. However, the values of CR

and TCR were within the range of $1 \times 10^{-4}$ and $1 \times 10^{-6}$ for adult men and adult women. This reflects the fact that intentional or non-intentional contact with soil, especially its ingestion, is hazardous for children in the region, but not for adults. The values of HI, THI, CR, and TCR in different population groups were in the order of children > adult females > adult males. It can thus be said that the potential risk of cancerous and non-cancerous diseases is higher for children than for adults. In the future, the simulation of the likely hazards of heavy metals by spatial and systemic analysis of these elements based on soil classification and the development of bio-clinical models for the whole soil–food–human system can contribute to reducing and managing the health risks of different population groups.

**Author Contributions:** S.R. and M.A. designed the study and wrote the manuscript and A.N. reviewed and edited the manuscript. All authors have read and agreed to the published version of the manuscript.

**Funding:** This research received no external funding.

**Institutional Review Board Statement:** Not applicable.

**Informed Consent Statement:** Not applicable.

**Data Availability Statement:** The data of the current study are available from the corresponding author upon a reasonable request.

**Acknowledgments:** The authors give a lot of thanks to Urmia University, Urmia, I.R. (Iran), for their support in this work.

**Conflicts of Interest:** The authors declare no conflict of interest.

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
