# Peer review of "Pollution Analysis and Health Implications of Heavy Metals under Different Urban Soil Types in a Semi-Arid Environment"

_sustainability, doi:10.3390/su151612157_

Round 1

Reviewer 1 Report

  1. Introduction

The introduction lacks a solid scientific basis for the studies conducted. The authors should briefly refer to the results of previous studies and not just mention that they were or were not conducted but emphasize what was relevant to the topic. In particular - the role of different soil types on the mobility of heavy metals, the role of different regions and soil types on the degree of contamination and danger of agricultural land.

The topic of the research undertaken was not sufficiently substantiated. That heavy metals are present in soils and their amount varies in different regions dependent on different pollutants is obvious. That the research should not be local in scope, the authors should focus more (according to the title of the article) on explaining what factors/processes affect the absorption capacity of metals in different types of soils. What are the differences in the behavior of metals in these specific types and what is known about this in the world literature (lines 73-74). As it is, the article more resembles a research report rather than a scientific article.

Lines 79-81 what came out of this research ?

Lines 93-95 Nonetheless, limited data are available on the extent of pollution and health risk of agricultural lands in urban and peri-urban areas located in arid and semi-arid regions and calcareous soils with diverse soil types . This has not been sufficiently proven in the introduction and why it is important how it may affect the results.

2 Material and methods

What was the basis for selecting the location of each profile? Justify

Fig.1 is not of good enough quality

Line 115 'Then, they were described, analyzed, and classified [19].' Just quoting is not enough add a few words of explanation so that the reader does not have to look for other sources.

From what depth were samples taken? How do you know that there are no embankment soils on the surface?

Lines 122 All methods should be described so that they can be reproduced even if a footnote is given. Addendum should be made.

The methods lack a description of the study of granulometric composition

Lines 157-163 Each formula should have its own separate legend to make it more readable

Table 1 - Is there any bibliography for Table 1 ?

Line 172 Data analysis should be described in more detail. Whether parametric or non-parametric tests were used, whether the data had a normal distribution. Why was Duncan's test chosen? What kind of correlations were performed (Pearson's linear or others?) and why? Were the data not correlated with each other?

3.Results and discussion

How do we know that the causes of the observed variability should be sought in the type of soil(genesis) and not in its physical and chemical properties ? Statistical analysis could help answer this question. With such a large variation in physicochemical properties in different soil types, the effect of changes in physicochemical properties should be taken in the 1st order than its genesis. Or genesis should be taken into account as an additional factor.

There should be a more in-depth analysis of the data. Correlations do not constitute a cause-and-effect relationship. If the conditions are met, it is worth conducting parametric tests of analysis of variance, post-hoc tests, an.regression.

Table 3 What is the composition of all elements in each soil type? The authors refer to the main metal cations below ?

Line 223 This should be better explained. Ca, Na, K cations do not allow absorption of metals ? This is not true. There are other factors that determine this.

The authors should complete the analyses of, for example, correlations with the main metals that are found in soils

Lines 234-235 What factor is involved?

Lines 3-4 Page 12 these methods should be described and justified in the data analysis section

The discussion should be supplemented with the probable causes of the observed variability. Data analysis alone has little scientific value.

4 Conclusions.

Conclusions should be improved. The authors should focus on drawing conclusions rather than summarizing the results of the study.

Author Response

Thank you for your detailed comments and attention. The comments made by you were respond as follows:

Reviewer # 1

Comment/ Question

Author' Response

  1. Introduction

The introduction lacks a solid scientific basis for the studies conducted. The authors should briefly refer to the results of previous studies and not just mention that they were or were not conducted but emphasize what was relevant to the topic. In particular - the role of different soil types on the mobility of heavy metals, the role of different regions and soil types on the degree of contamination and danger of agricultural land.

The topic of the research undertaken was not sufficiently substantiated. That heavy metals are present in soils and their amount varies in different regions dependent on different pollutants is obvious. That the research should not be local in scope, the authors should focus more (according to the title of the article) on explaining what factors/processes affect the absorption capacity of metals in different types of soils. What are the differences in the behavior of metals in these specific types and what is known about this in the world literature (lines 73-74). As it is, the article more resembles a research report rather than a scientific article.

Thanks for the suggestion, but we believe that the current introduction section fits to the journal guidelines which suggest that “Introduction should provide readers with the background information needed to understand your study, and the reasons why you conducted your experiments. The Introduction should answer the question: what question/problem did you study? We believe that Introduction with the current form fits well to journal guidelines

We also tried to avoid overwhelming the reader by following the journal guidelines which suggest that “introduction should provide readers with the information needed to understand your study, and the reasons why you conducted your study.

ü  Lines 79-81 what came out of this research

The output of such studies is shown in the upper paragraphs of the introduction (e.g., line 46 - 65).

ü  Lines 93-95 Nonetheless, limited data are available on the extent of pollution and health risk of agricultural lands in urban and peri-urban areas located in arid and semi-arid regions and calcareous soils with diverse soil types . This has not been sufficiently proven in the introduction and why it is important how it may affect the results.

Thanks for the suggestion, but we believe that sufficient background information about the heavy metal pollution in the agricultural soils of urban region and the associated health risk has been provided.

2 .Material and methods

ü  What was the basis for selecting the location of each profile? Justify

The landscape characteristics (e.g., elevation, slope, drainage), as well as soil variations and the target crop, were the parameters considered in the selection of soil profiles and the sampling locations

ü  Fig.1 is not of good enough quality

Fig. 1 was improved as suggested.

ü  Line 115 'Then, they were described, analyzed, and classified [19].' Just quoting is not enough add a few words of explanation so that the reader does not have to look for other sources.

The corrections were applied as suggested.

ü  From what depth were samples taken? How do you know that there are no embankment soils on the surface?

Thank you. Soil sampling depth was added.

Embankment effects were not observed in the field works.

ü  Lines 122 All methods should be described so that they can be reproduced even if a footnote is given. Addendum should be made.

In general, the format that we used to reflect the general soil analyses  is a common method that is followed in all similar articles published in international journals (e.g., Sustainability). Therefore, We believe that Laboratory Analyses with the current form fits well to journal guidelines.

ü  The methods lack a description of the study of granulometric composition

As mentioned in lines 126 - 127, soil particle size distribution is determined by the hydrometric method.

ü  Lines 157-163 Each formula should have its own separate legend to make it more readable.

The equations were formated based on the journal guidelines.

ü  Table 1 - Is there any bibliography for Table 1 ?

The symbols that are in table 1 were applied in the equations on pages 5 and 6 and their bibliography is presented there.

ü  Line 172 Data analysis should be described in more detail. Whether parametric or non-parametric tests were used, whether the data had a normal distribution. Why was Duncan's test chosen? What kind of correlations were performed (Pearson's linear or others?) and why? Were the data not correlated with each other?

Due the diversity of natural and human processes, some soil heavy metals (e.g., Cd and Pb) had not normal distribution. Duncan's test is a current method in soil science to statistical comparisons. Pearson's linear was used for assessing the  correlations between soil heavy metals and other soil parameters. Some correction was carried out in  Data Analysis as suggested.

3.Results and discussion

Thank you for your comment. Corrected as suggested. Page of 10, line 274-275.

ü  How do we know that the causes of the observed variability should be sought in the type of soil(genesis) and not in its physical and chemical properties ? Statistical analysis could help answer this question. With such a large variation in physicochemical properties in different soil types, the effect of changes in physicochemical properties should be taken in the 1st order than its genesis. Or genesis should be taken into account as an additional factor.

There should be a more in-depth analysis of the data. Correlations do not constitute a cause-and-effect relationship. If the conditions are met, it is worth conducting parametric tests of analysis of variance, post-hoc tests, an.regression.

As stated in the different sections of the results and discussion, we believe that a combination of changes in soil formation processes, physical and chemical properties and anthropogenic proccesess have caused changes in the characteristics of the study soils and their heavy metals. These reasons have been documented in detail in previous studies conducted in this area.

·       Rezapour, S., Samadi, A. and Khodaverdiloo, H., 2012. Impact of long-term wastewater irrigation on variability of soil attributes along a landscape in semi-arid region of Iran. Environmental Earth Sciences67, pp.1713-1723.

·       Rezapour, S. and Moazzeni, H., 2016. Assessment of the selected trace metals in relation to long-term agricultural practices and landscape properties. International Journal of Environmental Science and Technology13, pp.2939-2950.

·       Mamehpour, N., Rezapour, S. and Ghaemian, N., 2021. Quantitative assessment of soil quality indices for urban croplands in a calcareous semi-arid ecosystem. Geoderma382, p.114781.

ü  Table 3 What is the composition of all elements in each soil type? The authors refer to the main metal cations below ?

Table 3 just showed summary statistics of heavy metals concentration of the soils for different soil types along with statistical comparison the mean valuse of heavy metals between four soil types were studied. The Tabe shows that the mean concention of the majority of metals was significantly higher in Fluvisols than other soil types.

ü  Line 223 This should be better explained. Ca, Na, K cations do not allow absorption of metals ? This is not true. There are other factors that determine this.

Corrections was added.

ü  The authors should complete the analyses of, for example, correlations with the main metals that are found in soils

Correlations between heavy metals with main soil attributes showed in Table 4

ü  Lines 234-235 What factor is involved?

Corrected.

ü  Lines 3-4 Page 12 these methods should be described and justified in the data analysis section

Corrected.

ü  The discussion should be supplemented with the probable causes of the observed variability. Data analysis alone has little scientific value.

We attempted to describe the results in a way that answered the research questions, highlighted the most important results, and linked our results to previous research and theory.

We wish the reviewer could provide us with more specific comments about the revisions required for the statement about the implications of result.

ü  Conclusions should be improved. The authors should focus on drawing conclusions rather than summarizing the results of the study.

Thank you for your comment, but we prefer to stick to the journal guidelines which suggest that conclusion should provide readers with primary viewpoint, shortcomings, academic contribution, and some suggestion for future research.

Reviewer 2 Report

The manuscript is in generall well written. However, the number of sampling points is limited (only 10). I suggest the authors to clearly inform the readers that the results from these samples do not represent the risk condition of the study area, due to limited sample numbers and the high variability of soil  heavy metal contents.

Author Response

Thank you for your detailed comments and attention. The comments made by you were respond as follows:

Reviewer # 2

ü  The manuscript is in generall well written. However, the number of sampling points is limited (only 10). I suggest the authors to clearly inform the readers that the results from these samples do not represent the risk condition of the study area, due to limited sample numbers and the high variability of soil  heavy metal contents.

As showed in line 114 -123,  10 soil profiles were allocated in a urban area. Our primary goal was to include as much spatial variabilities as possible in the sampling scheme. Each soil profile was considered a central sampling location and four additional composite (across depth) soil samples were collected at four cardinal directions each being 50–100 m from the central point (soil profile) (n = 50). Therefor, the number of sampling points was 50 points.

Reviewer 3 Report

Rewiev

The work "pollutionsisis and health implications of heavy metals  under diffferent urban soil types" is interesting concerns the effects of soil pollution. However, I think that you need to change the title because the research also applies to agricultural soils located outside the city.

After analyzing, I have the following comments, which is taken into account, which is a condition for admission to the further stage of the procedure. My comments and suggestions have been applied in the form of comments in the PDF file.

Author Response

Thank you for your detailed comments and attention. The comments made by you were respond as follows:

Reviewer # 3

ü  The work "pollutionsisis and health implications of heavy metals under diffferent urban soil types" is interesting concerns the effects of soil pollution. However, I think that you need to change the title because the research also applies to agricultural soils located outside the city.

Thank you for your kind comments. We tried that the title of our manuscript be specific as possible while still describing the full range of the study. However, if the reviewer has any suggestions for title of the manuscript, we welcome it.

ü  After analyzing, I have the following comments, which is taken into account, which is a condition for admission to the further stage of the procedure. My comments and suggestions have been applied in the form of comments in the PDF file.

Thank you. Corrections was carried out as suggested.

Round 2

Reviewer 1 Report

Introduction should provide readers with the background information needed to understand your study, and the reasons why you conducted your experiments. The Introduction should answer the question: what question/problem did you study? We believe that Introduction with the current form fits well to journal guidelines

It is not true. Acc. Sustainabillity authors guidelines:

Introduction ‘The introduction should briefly place the study in a broad context and highlight why it is important. It should define the purpose of the work and its significance. The current state of the research field should be carefully reviewed and key publications cited. Please highlight controversial and diverging hypotheses when necessary. Finally, briefly mention the main aim of the work and highlight the principal conclusions. As far as possible, please keep the introduction comprehensible to scientists outside your particular field of research.’

The introduction of the study lacks a robust scientific foundation, specifically concerning the role of different soil types in relation to the degree of contamination and the potential risks posed to agricultural land. Furthermore, the significance of addressing this topic and its relevance to the broader community, beyond local implications, is inadequately justified

In general, the format that we used to reflect the general soil analyses is a common method that is followed in all similar articles published in international journals (e.g., Sustainability). Therefore, We believe that Laboratory Analyses with the current form fits well to journal guidelines

Methods’ The Materials and Methods should be described with sufficient details to allow others to replicate and build on the published results. Please note that the publication of your manuscript implicates that you must make all materials, data, computer code, and protocols associated with the publication available to readers. Please disclose at the submission stage any restrictions on the availability of materials or information. New methods and protocols should be described in detail while well-established methods can be briefly described and appropriately cited.’

All methods should be described so that they can be reproduced even if a footnote is given. 

Due the diversity of natural and human processes, some soil heavy metals (e.g., Cd and Pb) had not normal distribution. Duncan's test is a current method in soil science to statistical comparisons. Pearson's linear was used for assessing the correlations between soil heavy metals and other soil parameters. Some correction was carried out in Data Analysis as suggested.

The authors need to provide a justification for their selection of statistical tools, specifically explaining why they chose parametric tests such as analysis of variance, Pearson's R coefficient, and Duncan's test, despite the data not meeting the assumptions required for parametric tests, including the assumption of a normal distribution. When the assumptions for parametric tests are not met, the basis for drawing inferences from them becomes incorrect, and the credibility of the study is compromised. Additionally, the results of the Kaiser-Meyer-Olkin test, analysis of variance, and the Duncan test, as mentioned in the article, should be presented. It is also essential to include the number 'N' next to each table, which represents the sample size.

Thank you for your comment, but we prefer to stick to the journal guidelines which suggest that conclusion should provide readers with primary viewpoint, shortcomings, academic contribution, and some suggestion for future research.

There is a lack of information on scientific contributions and shortcomings in the conclusions

Author Response

We would like to thank from you for the thoughtful comments on our manuscript. The comments made by you were respond as follows:

Reviewer # 1

Comment/ Question

Author' Response

Introduction

The introduction of the study lacks a robust scientific foundation, specifically concerning the role of different soil types in relation to the degree of contamination and the potential risks posed to agricultural land. Furthermore, the significance of addressing this topic and its relevance to the broader community, beyond local implications, is inadequately justified

We believe that the introduction has sufficiently revised in the first revision (line 35 - 53 and 83- 102) and there is no need for further revisions.

More, there is not the standard reference which showed the role of different soil types in relation to the degree of contamination. if the reviewer has any references for this regard, we welcome it.

Methods

All methods should be described so that they can be reproduced even if a footnote is given. 

Laboratory Analyses was revised as possible.

The authors need to provide a justification for their selection of statistical tools, specifically explaining why they chose parametric tests such as analysis of variance, Pearson's R coefficient, and Duncan's test, despite the data not meeting the assumptions required for parametric tests, including the assumption of a normal distribution. When the assumptions for parametric tests are not met, the basis for drawing inferences from them becomes incorrect, and the credibility of the study is compromised.

As mentioned in the first revision, our data had a normal distribution, except for the value of Cd and Pb in some soil profiles, which were also normalized by standard methods (Reimann et al., 2011). Accordingly, we used the parametric tests for data analysis.

Reimann, C., Filzmoser, P., Garrett, R. and Dutter, R., 2011. Statistical data analysis explained: applied environmental statistics with R. John Wiley & Sons.

Additionally, the results of the Kaiser-Meyer-Olkin test, analysis of variance, and the Duncan test, as mentioned in the article, should be presented. It is also essential to include the number 'N' next to each table, which represents the sample size

Corrected.

There is a lack of information on scientific contributions and shortcomings in the conclusions

We think the conclusion is adequate. Please refer to lines of 64 – 65 and 68 – 76 (in conclusion section) for scientific contributions and shortcomings.

Reviewer 3 Report

Review

Unfortunately, not all my suggestions were taken into account, moreover, I found differences in the results in the current version compared to the old one.

Also the subject, as I wrote earlier, should be changed, because it misleads the reader. The studied soil profile are mostly located in suburban areas, so the topic should be changed, e.g.: Pollution Analysis and Health Implications of Heavy Metals under Different Soil Types in the upper levels of soil profiles located in the Urmia City suburban area. (Western Azerbaijan Province).

I asked too for a short description along with the basic physical and chemical properties of the soil profiles, where they were placed.

In Table 2, why these results have changed from the first version.

In Table 2, are the results given as percentage of organic carbon or organic matter? The organic matter results express the percentage of organic carbon or organic matter and that is the difference.

Author Response

We would like to thank from you for the thoughtful comments on our manuscript. The comments made by you were respond as follows:

Reviewer # 3

ü  As I wrote earlier, should be changed, because it misleads the reader. The studied soil profile is mostly located in suburban areas, so the topic should be changed, e.g.: Pollution Analysis and Health Implications of Heavy Metals under Different Soil Types in the upper levels of soil profiles located in the Urmia City suburban area. (Western Azerbaijan Province).

We tried that the title of our manuscript be concise, specific and relevant as suggested by MDPI. However, our changed the title of manuscript as "Pollution Analysis and Health Implications of Heavy Metals under Different Urban Soil Types in a Semi-Arid Environment"

ü  I asked too for a short description along with the basic physical and chemical properties of the soil profiles, where they were placed.

Some corrections were add (line 121-125 and 130-131) regarding “a short description along with the basic physical and chemical properties of the soil profiles”. Basic physical and chemical properties of the soils are described in 3.1 section.

ü  In Table 2, why these results have changed from the first version.

You right. This manuscript is extracted from my master's student thesis. in this regard, some of the data were misplaced by my student and some of them were also mistaken compared to the original data. Therefore, we modified this data based on the original data.

ü  In Table 2, are the results given as percentage of organic carbon or organic matter? The organic matter results express the percentage of organic carbon or organic matter and that is the difference.

In Table, OM is percentage of organic matter. Please refer to the Acronyms/Abbreviations of Table 2.

Round 3

Reviewer 1 Report

The authors made no changes to the introduction regarding the analysis of metal occurrence in each soil type. This supplement could have enhanced the scientific value of the paper. As the authors rightfully pointed out, there is still much to be discovered on this topic. The health-related aspect that the authors focused on is extensively covered here and appears sufficient for publication.

The methods are briefly supplemented, yet they are adequate. Please add the number of the standard according to which the test was carried out at Line 136.

Avoid using the personal form at Line 193.

For Table 2 and Table 3, include explanations under the tables for SD (Standard Deviation) and CV (Coefficient of Variation). Additionally, next to the SD values, it is essential to add the symbol "±" to indicate the confidence intervals.

Please clarify the number of observations (N) for each soil type below the table. It is not entirely clear where N=50 came from. 4 soil types, 3 replications from each. This gives 12 samples. On the other hand, there were 10 profiles. This aspect needs to be elucidated in the article to avoid confusion.

Note that N represents the number of observations of a given type, not the number of samples. Thus, N will not be equal in each table.

At Line 285, use "p" rather than "P" to refer to statistical significance.

In Fig. 2, please provide a clear explanation of the difference between the letters "a" and "b," which represent confidence intervals. This distinction should be made evident.

Furthermore, in other figures, the symbols "a" and "b" appear to have different meanings or are repeated (e.g., Fig. 3). It is essential to standardize and clarify the meaning of these symbols in all figures to ensure consistency throughout the article.

Please correct any editing errors found throughout the manuscript, including issues like the font in tables or others (e.g., Line 142).

Author Response

Dear reviewer

We would like to thank again from you for the thoughtful positive comments on our manuscript. The comments made by you were respond as follows and highlighted by blue mark in text:

Comment/ Question

Author' Response

The methods are briefly supplemented, yet they are adequate. Please add the number of the standard according to which the test was carried out at Line 136.

Corrected (page of 4, line 137).

Avoid using the personal form at Line 193.

Corrected (page of 6, line 192).

For Table 2 and Table 3, include explanations under the tables for SD (Standard Deviation) and CV (Coefficient of Variation). Additionally, next to the SD values, it is essential to add the symbol "±" to indicate the confidence intervals.

Corrected (page of 6-7 and 9, Tables of 2 and 3).

Please clarify the number of observations (N) for each soil type below the table. It is not entirely clear where N=50 came from. 4 soil types, 3 replications from each. This gives 12 samples. On the other hand, there were 10 profiles. This aspect needs to be elucidated in the article to avoid confusion.

Corrected (page of 6-7 and 9, Tables of 2 and 3).

At Line 285, use "p" rather than "P" to refer to statistical significance.

Corrected (page of 10, line 292).

In Fig. 2, please provide a clear explanation of the difference between the letters "a" and "b," which represent confidence intervals. This distinction should be made evident.

Corrected (title of Figure 2).

Furthermore, in other figures, the symbols "a" and "b" appear to have different meanings or are repeated (e.g., Fig. 3). It is essential to standardize and clarify the meaning of these symbols in all figures to ensure consistency throughout the article.

Corrected (title of Figures 3 and 4).

Please correct any editing errors found throughout the manuscript, including issues like the font in tables or others (e.g., Line 142).

Corrected (all Tables).